# Hyperspectral Estimation of Winter Wheat Leaf Area Index Based on Continuous Wavelet Transform and Fractional Order Differentiation

**DOI:** 10.3390/s21248497

**Published:** 2021-12-20

**Authors:** Changchun Li, Yilin Wang, Chunyan Ma, Fan Ding, Yacong Li, Weinan Chen, Jingbo Li, Zhen Xiao

**Affiliations:** 1School of Surveying and Land Information Engineering, Henan Polytechnic University, Jiaozuo 454000, China; lichangchun610@126.com (C.L.); 211904010019@home.hpu.edu.cn (Y.W.); 212004020067@home.hpu.edu.cn (F.D.); 211904020026@home.hpu.edu.cn (Y.L.); 211904020046@home.hpu.edu.cn (W.C.); lijingbo1024@163.com (J.L.); 212004020068@home.hpu.edu.cn (Z.X.); 2Beijing Research Center for Information Technology in Agriculture, Beijing Academy of Agriculture and Forestry Sciences, Beijing 100097, China

**Keywords:** winter wheat, leaf area index, fractional order differential, continuous wavelet transform, optimal subset regression, support vector machine

## Abstract

Leaf area index (LAI) is highly related to crop growth, and the traditional LAI measurement methods are field destructive and unable to be acquired by large-scale, continuous, and real-time means. In this study, fractional order differential and continuous wavelet transform were used to process the canopy hyperspectral reflectance data of winter wheat, the fractional order differential spectral bands and wavelet energy coefficients with more sensitive to LAI changes were screened by correlation analysis, and the optimal subset regression and support vector machine were used to construct the LAI estimation models for different growth stages. The precision evaluation results showed that the LAI estimation models constructed by using wavelet energy coefficients combined with a support vector machine at the jointing stage, fractional order differential combined with support vector machine at the booting stage, and wavelet energy coefficients combined with optimal subset regression at the flowering and filling stages had the best prediction performance. Among these, both flowering and filling stages could be used as the best growth stages for LAI estimation with modeling and validation *R*^2^ of 0.87 and 0.71, 0.84 and 0.77, respectively. This study can provide technical reference for LAI estimation of crops based on remote sensing technology.

## 1. Introduction

Leaf area index (LAI) is one of the important community structure parameters in ecosystem research, directly related to many ecological processes such as evapotranspiration, soil water balance, and net productivity. In addition, it is also an important spatial variable measuring wheat photosynthetic effective radiation, transmission, and an eco-environmental process model [1]. In addition, LAI is an important characteristic parameter describing the geometric structure of the wheat canopy. It can be used to quantitatively express the initial energy exchange on the canopy surface, directly reflect the energy, carbon dioxide, and physical environment of growth in the canopy diversification scale space, and reflect the dynamic characteristics and health status of wheat in the process of growth and development. Therefore, LAI estimation is essential for wheat growth monitoring and yield estimation. In the traditional LAI acquisition method, measurement was conducted on the field, which turns out to be destructive, time-consuming, and laborious, and not able to obtain LAI continuously in real time and on a large scale. Remote sensing has the factors of high temporal and spatial resolution and can be used to monitor a region quickly, widely, and periodically, which has now become the main means to estimate surface parameters and extract crop phenotypic parameters.

At present, many scholars have conducted research on crop LAI estimation using remote sensing technology and made some achievements. For example, Huang Jingfeng et al., Su Wei et al., and Fieuzal R et al. [2,3,4] calculated the position, amplitude, and amplitude of the red edge, respectively, by using the C _HH_ and L _HH_ band data of canopy red light band (680~760 nm), extracted lidar, and SAR parameters such as red edge area and vertical structure parameters, analyzed their correlation with LAI, and constructed LAI estimation models of rape, corn, and wheat. Liu Jun et al., Li Shumin et al., and Wang Laigang et al. [5,6,7] established LAI estimation models of maize and wheat at different growth stages by constructing a vegetation index and a spectral index, combining with measured LAI data and satellite remote sensing data, such as environmental satellite, MODIS, aster, and SPOT5, using mathematical statistics or the crop growth model PROSAIL, so as to realize LAI dynamic monitoring. By analyzing the current research situation, it could be seen that most of the existing studies are based on the original spectrum. They used the original spectrum to construct the correlation index, analyzed its relationship with crop LAI, and constructed the crop LAI remote sensing estimation model. Spectral differentiation technique can reflect the essential characteristics of crops by partially eliminating the influence of environment, such as atmospheric effect, vegetation shadow, and soil. In recent years, more and more attention has been paid to the monitoring of crop growth by spectral derivative technique and some research results have been obtained. For example, Wang Dengwei et al. [8] found that the first-order differential spectrum value of 750 nm is highly correlated with chlorophyll content. Chen Junying et al. [9] analyzed the characteristics of chlorophyll content and spectrum of rice and found that the first-order differential parameter of spectral reflectance of rice leaves has a strong correlation with chlorophyll content. Jiang Jinbao et al. [10,11,12] analyzed the correlation between different order spectral derivative indexes and wheat stripe rust and canopy chlorophyll content, optimized the spectral derivative indexes with a strong correlation, and constructed a wheat stripe rust monitoring model and canopy chlorophyll content estimation model, which achieved good results. Smith K L et al. [13] showed that the first-order differential ratio at 725 and 702 nm can be used to monitor the vegetation growth under gas leakage stress. Fractional order differentiation, as the generalization of integer order differentiation, was first proposed by Leibniz at the end of the 17th century and laid some theoretical foundation for its future development [14]. Fractional order differential techniques have unique advantages in signal analysis and processing and have been widely used recently in biological fields, physics, and engineering, among others [15,16]. In order to make full use of the spectral information acquired by the remote sensing platform, enhance the effectiveness of spectral response, and improve the accuracy of spectral modeling, it is necessary to preprocess the raw spectral data. Most previous studies have used integer-order differential transformations to process spectral data, which may cause partial loss of spectral information. The use of fractional order differential transform can realize the refinement of spectral information, which can effectively eliminate the influence of environmental background and deeply explore the potential information in the spectrum [17,18]. Jiang Ming et al. [19] studied the correlation between 0~2 order (interval 0.2 order) fractional differential spectral reflectance and heavy metal content in soil, obtained the correlation coefficients of each order fractional differential and soil heavy metals at each spectral sampling interval, and compared and analyzed the curve change law of correlation coefficients. Zhang Wenwen et al. [20] processed the spectral reflectance with the 11-order differential of 0~1 order (with an interval of 0.1 order) and analyzed the correlation between the differential value of each order and the measured Cu~(2+) content in corn leaves. The results showed that, compared with the common first-order differentiation, the fractional differentiation can highlight the correlation between the spectral reflectance of some bands and the Cu~(2+) content in leaves and expand the selection space of the characteristic band. Wang Jingzhe et al. [21] discussed the possibility of fractional differentiation technology in estimating the content of heavy metal chromium and organic carbon in desert soil using hyperspectral data. The research results showed that the accuracy and robustness of the model after fractional preprocessing are higher than that of integer differentiation. Li Changchun et al. [22] processed hyperspectral remote sensing data by factional differentiation, analyzed the correlation between fractional differentiation spectrum and wheat chlorophyll content, and constructed a chlorophyll content estimation model by using stepwise regression analysis, SVM, and artificial neural network. Spectral signal transformation can improve its sensitivity to crop LAI. Continuous wavelet transform (CWT) can effectively reduce noise, decompose spectral data, and extract more spectral positions and characteristic parameters [23]. At present, the wavelet coefficients extracted by CWT are effective in the inversion of heavy metals and phenotypic parameters in the crop canopy. For example, Chen Haoyu et al. [24] performed CWT on hyperspectral data to generate wavelet energy coefficient and established a BP neural network and SVM inversion model of soil organic matter content. Li Bao et al. [25] and Tan Xianming et al. [26] used univariate linear regression, SVM, and the partial least square method to construct the chlorophyll content estimation model for fresh peach leaves and maize canopy after continuous wavelet transform of spectral information, which was better than the traditional methods. Wang Yancang et al. [27] carried out CWT on hyperspectral data and constructed a quantitative inversion model of winter wheat leaf water content by using partial least square method.

This paper aims to construct and screen the best estimation models of LAI for different growth stages, with winter wheat as the research object. This study, firstly, preprocessed the canopy hyperspectral data by fractional order differential transform and continuous wavelet decomposition. Then, the fractional order differential spectral bands and wavelet energy coefficients with stronger correlation with the LAI based on correlation analysis were screened. Finally, the LAI estimation models for jointing, booting, flowering, and filling stages were constructed and evaluated by support vector machine and optimal subset regression, respectively. This study can provide theoretical and technical references for remote sensing estimation of crop LAI.

## 2. Materials and Methods

### 2.1. Overview of the Study Area and Experimental Design Scheme

The research area is located in the National Precision Agriculture Research and Demonstration Base in Xiaotangshan Town, Changping District, Beijing. It is flat and fertile tidal soil. The average altitude is approximately 36 m. The climate is mild with four distinct seasons, the average annual temperature is approximately 13 °C, the average annual rainfall is approximately 510 mm, and precipitation is mostly concentrated in summer, characterizing a typical warm temperate continental monsoon climate. Figure 1 shows the specific location of the study area.

The experimental area was 84 m in length from east to west and 32 m in length from north to south, with each plot measuring 6 × 8 m. There were 48 plots, 16 treatments, and 3 replications. The orthogonal experiment was conducted with different amounts of nitrogen fertilizer, different moisture contents, and different crop varieties. Four levels of *N* fertilizer were set: 0, 195, 390, and 585 kg/hm^2^, respectively; three levels of water irrigation were set: rainfed (W1, no irrigation), normal water (W2, irrigation water 192 mm), and twice normal water (W3, irrigation water 384 mm); there are two crop varieties: Jing 9843 (J9843) and Zhong Mai 175 (ZM175). The sowing time was 7 October 2014, the planting density was 3.75 million plants/hm^2^, the harvest date was on 11 June 2015, and the crops before sowing were corn.

### 2.2. Data Acquisition

Canopy hyperspectral data and LAI data were measured at the jointing stage, booting stage, flowering stage, and filling stage of winter wheat in 2018 and 2019, respectively.

#### 2.2.1. Canopy Hyperspectral Data Measurement

The canopy hyperspectral data were measured using an ASD Field SpecFR Pro 2500 spectrometer. The data collection was conducted during 10:00–14:00 BST on a clear day, with a sensor probe field of view of 25°, and placed vertically downward at approximately 1.0 m above the canopy. To eliminate the effect of environmental changes on the spectral measurements, the spectrometer was calibrated with a whiteboard before and after each measurement, and dark currents were collected every 5 min. Measurements were repeated 10 times on each plot and the average value was taken as the spectral reflectance of the canopy on the plot. Since the spectrometer has different spectral sampling intervals in different bands, after the spectral data acquisition, the spectral resampling interval was first set to 1 nm, and then the spectrum was smoothed using the Savitzky–Golay filter [28] of ViewSpec Pro software (Malvern Panalytical, Malvern, UK) to reduce the spectral noise and improve the signal-to-noise ratio of the spectral data.

#### 2.2.2. LAI Data Measurement

LAI was measured using the LAI 2200 plant canopy analyzer. For wheat LAI measurement, three sample points were randomly selected for each plot. The LAI 2200 first performed backlit measurement in an open area to ensure that all LAI data obtained were accurate and valid. The LAI 2200 was then placed at the sample point parallel to the ridge and perpendicular to the ridge, respectively, with the probe placed close to the wheat plant. LAI values were measured 4 times, and the arithmetic mean value was taken as the LAI of the sample point and then the average of the LAI values at the three sample points was taken as the LAI value of the plot.

### 2.3. Data Processing Methods

#### 2.3.1. Fractional Order Derivative Processing

Fractional order derivative is a fundamental mathematical operation with a wide range of applications in fields such as image enhancement processing and signal analysis [29,30]. Traditional integer order derivative will ignore some information related to crop physiological and biochemical parameters, which affects the accuracy, while fractional order derivative can effectively denoise and refine the local information of hyperspectral data to obtain more detailed information. The commonly used fractional order derivative includes three types as follows: Riemann–Liouville, Caputo, and Grünwald–Letnikov [31]. The paper uses the Grünwald–Letnikov differentiation to process canopy hyperspectral data. The differential formula is as shown in Equation (1):(1)$$\frac{d^{\alpha }f(\lambda )}{d\lambda^{\alpha }}=f(\lambda )+(-\alpha )f(\lambda -1)+\frac{\left(-\alpha )(-\alpha +1\right)}{2}f(\lambda -2)+\cdots +\frac{\Gamma (-\alpha +1)}{m!\Gamma (-\alpha +1)}f(\lambda -m)$$

In the formula, Γ(⋅) is the Gamma function, λ is the corresponding wavelength, m represents the difference between the upper and lower limits of the differential, α represents any order.

#### 2.3.2. Continuous Wavelet Transform

Wavelet transform is called “mathematical microscope”, which can decompose complex signals into wavelet signals of different scales (frequencies), with rich basis functions, good time-frequency localization, and multi-scale characteristics. There are two main types of wavelet transforms, that is, continuous wavelet transform (CWT) and discrete wavelet transform (DWT). The paper used CWT to decompose the hyperspectral data in order to obtain a series of wavelet energy coefficients at different scales. The wavelet coefficients contain two dimensions of decomposition scale and band. Therefore, the 1D hyperspectral data are converted to 2D wavelet energy coefficients by continuous wavelet transform. The calculation is shown in Equation (2):(2)$$W_{f}(a,b)=\int_{-\infty }^{+\infty }f(\lambda )\Psi_{a,b}(\lambda )d\lambda$$
(3)$$\Psi_{a,b}(\lambda )=\frac{1}{\sqrt{a}}\Psi (\frac{\lambda -b}{a})$$
where f(λ) is the hyperspectral reflectance, λ is the spectral band in the range of 350–1350 nm, $\Psi_{a,b}$ denotes the wavelet basis function, a denotes the scale factor, b is the translation factor.

### 2.4. Modeling Methods

#### 2.4.1. Optimal Subset Regression

Optimal subset regression is a method combining all alternative independent variables as a subset of the model for regression modeling. For a model with independent variables, the optimal subset can be used to build $2^{n-1}$ subset models, and the selection is performed to determine the best combination of independent variables for the model by using the maximum adjusted $R^{2}$ ($Adj\cdot R^{2}$) as the principle of independent variable selection [32].
(4)$$Adj\cdot R^{2}=1-\frac{RSS/(n-k-1)}{TSS/(n-1)}$$
(5)$$RSS=\sum_{i=1}^{n}{\left(y_{i}-{\overset{⌢}{y}}_{i}\right)}^{2}$$
(6)$$TSS=\sum_{i=1}^{n}{\left(y_{i}-\bar{y_{i}}\right)}^{2}$$
where RSS denotes the residual sum of squares, TSS denotes total sum of square difference, $y_{i}$ is the measured value, ${\overset{⌢}{y}}_{i}$ is the estimated value, $\bar{y_{i}}$ is the estimated value of the measured value, n is the number of samples, k is the number of independent variables, and i is the sample identifier.

#### 2.4.2. Support Vector Machines

Support vector machine (SVM) is a machine learning algorithm based on supervised learning and the principle of structural risk minimization. By projecting data into a high-dimensional space through a kernel function and finding the optimal hyperplane in the high-dimensional space, it better solves the dimensional catastrophe and overfitting problems, with good generalization ability and robustness. Therefore, it is usually used for pattern recognition, classification, and small sample regression analysis [33]. Support vector machines are more stable in training and are capable of obtaining more accurate results when used for small sample regression analysis. So, it can be used to find the optimum directly in the learning performance and complexity of the model based on limited data information, with a view to obtaining the best generalization capability.

### 2.5. Model Accuracy Evaluation

In the paper, 75% of the sample data was selected for model construction and the rest was used for model accuracy validation. The coefficient of determination ($R^{2}$), root mean squared error (RMSE), and standard root mean squared error (nRMSE) are selected as model accuracy evaluation indexes, and each evaluation indexes are calculated as follows.
(7)$$R^{2}=\frac{(\sum_{i=1}^{n}y_{i}-\bar{y}{)}^{2}}{(\sum_{i=1}^{n}x_{i}-\bar{y}{)}^{2}}$$
(8)$$RMSE=\sqrt{\frac{\sum_{i=1,j=1}^{n}{\left(x_{i}-y_{i}\right)}^{2}}{n}}$$
(9)$$nRMSE=\sqrt{\frac{\sum_{i=1,j=1}^{n}{\left(x_{i}-y_{i}\right)}^{2}}{n}}/\bar{y}$$
where $x_{i}$ is the measured value, $y_{i}$ is the estimated value, $\bar{y}$ is the mean value, i is the sample identifier, and n is the number of samples.

In general, a larger $R^{2}$ and a smaller *RMSE* indicate better modeling results. In addition, nRMSE≤10% indicates that the consistency between measured and estimated values is excellent, 10%<nRMSE≤20% indicates that the consistency between measured and estimated values is good, 20%<nRMSE≤30% indicates that the consistency between measured and estimated values is moderate, and nRMSE>30% indicates that the consistency between measured and estimated values is poor.

## 3. Results

### 3.1. Estimation of Wheat LAI Based on Fractional Order Differential Spectra

#### 3.1.1. Correlation Analysis of Raw Spectra and Fractional Order Differential Spectra with Wheat LAI

The fractional order differentiation of the original hyperspectral data was performed using the Grünwald–Letnikov differentiation with order range 0–2 and step size 0.1, when the original spectrum, the first order differentiation spectrum, and the second order differentiation spectrum are represented. Twenty fractional order differential transformations were applied to the canopy raw spectra at the jointing stage, booting stage, flowering stage, and filling stage, respectively. The correlations between the raw spectra and differential spectra of each order and LAI were plotted for different fertility periods, and the results are shown in Figure 2 and Figure 3. Meanwhile, the 10 differential spectra with a strong correlation were selected and plotted with the LAI correlation matrix, and the results are shown in Figure 4.

At the jointing stage, Figure 2a, using raw spectral reflectance with LAI to analyze correlatively, showed that it was significantly negatively correlated with LAI at the 0.01 level in the band range of 350–716 nm and significantly positively correlated with LAI at the 0.01 level in the band range of 733–1318 nm, with a maximum absolute value of the correlation coefficient |ρ| = 0.72. Correlation analysis using fractional order differential spectra and LAI and analysis of Figure 3a showed that the maximum value of the absolute value of the correlation coefficient |ρ| between each order of differential spectra and LAI was above 0.72 and, when the order was 1, the maximum value of |ρ| was up to 0.77. The number of spectral bands passing the 0.01 highly significant level test was above 946 except for the integer orders (1st and 2nd orders) and was up to 977 when the order was 1.1. The orders and bands where the 10 differential spectra with high correlation coefficients were located were, respectively, 1st order, 1281 nm, 2nd order, 708 nm, 1st order, 757 nm, 1st order, 956 nm, 1.1 order, 708 nm, 1st order, 492 nm, 1.2 order, 702 nm, 1.3 order, 700 nm, 1.9 order, 696 nm, 1.4 order, 697 nm, and their correlation with the correlation matrix of LAI was shown in Figure 4a.

At the booting stage, Figure 2b, using raw spectral reflectance with LAI to analyze correlatively, indicated it was significantly negatively correlated with LAI at the 0.01 level in the band range of 350–723 nm and significantly positively correlated with LAI at the 0.01 level in the band range of 741–1141 nm, with the maximum absolute value of the correlation coefficient |ρ| = 0.78. The correlation between fractional order differential spectra and LAI was analyzed. Figure 3b showed that the maximum absolute value of the correlation coefficient |ρ| between each order of differential spectra and LAI was above 0.78 and, when the order was 1, the maximum value of |ρ| could reach 0.83. Except for the integer order (1st and 2nd order), the number of spectral bands passing the 0.01 highly significant level test was above 738, and when the order was 0, 0.1, 0.2, 0.3, 0.4, and 0.5, the maximum number of bands was up to 775. The orders and bands where the 10 differential spectra with high correlation coefficients were located were as follows: 1st order, 756 nm, 1st order, 1140 nm, 2nd order, 752 nm, 1st order, 499 nm, 1.9 order, 654 nm, 1.1 order, 711 nm, 1.8 order, 654 nm, 1.2 order, 654 nm, 1.7 order, 654 nm, 0.9 order, 621 nm, and their correlation with the correlation matrix of LAI was shown in Figure 4b.

At the flowering stage, Figure 2c, using raw spectral reflectance with LAI to analyze correlatively, said that it was significantly negatively correlated with LAI at the 0.01 level in the band range of 350–723 nm and significantly positively correlated with LAI at the 0.01 level in the band range of 736–1154 nm, with the maximum absolute value of the correlation coefficient |ρ| = 0.79. The correlation between the fractional order differential spectra and LAI was analyzed. It was showed in Figure 3c that the maximum value of the absolute value of the correlation coefficient |ρ| between each order of differential spectra and LAI was above 0.79 and, when the order was 1, the maximum value of |ρ| could reach 0.88. Except for the integer order (1st and 2nd order), the number of spectral bands passing the 0.01 highly significant level test was above 757 and when the order was 0, 0.1, 0.2, 0.3, 0.4, the maximum number of bands could be 793. The orders and bands where the 10 differential spectra with high correlation coefficients were located were as follows: 1st order, 754 nm, 1st order, 1127 nm, 1st order, 1309 nm, 2nd order, 762 nm, 1.1 order, 904 nm, 1.9 order, 867 nm, 1.2 order, 903 nm, 1.8 order, 903 nm, 0.9 order, 866 nm, 1.8 order, 820 nm, and their correlation matrix with LAI was shown in Figure 4c.

At the filling stage, Figure 2d, using raw spectral reflectance with LAI to analyze correlatively, showed that it is significantly negatively correlated with LAI at the 0.01 level in the 350–727 and 1320–1350 nm band ranges and significantly positively correlated with LAI at the 0.01 level in the 736–1144 nm band range, with the maximum absolute value of the correlation coefficient |ρ| = 0.83. The correlation between fractional order differential spectra and LAI was analyzed. Figure 2d showed that the maximum value of the absolute value of the correlation coefficient |ρ| between each order differential spectrum and LAI was above 0.72. When the order was 1, |ρ| can reach up to 0.82. The number of spectral bands passing the 0.01 highly significant level test was above 766, except for integer orders (1st and 2nd orders), and up to 794 when the order was 0 and 0.1. The 10 differential spectra with high correlation coefficients were located in the order and band were as follows: 1st order, 498 nm, 1st order, 687 nm, 1st order, 538 nm, 1st order, 752 nm, 1st order, 1152 nm, 2nd order, 753 nm, 1st order, 730 nm, 1st order, 367 nm, 1st order, 558 nm, 1st order, 584 nm, and their correlation with LAI correlation matrix was shown in Figure 4d.

#### 3.1.2. Construction of LAI Estimation Model Based on Optimal Subset Regression

During the construction of the LAI estimation model, it can be seen from the results of the optimal subset analysis (shown in Figure 5) that, at the jointing stage, five fractional order differential spectra, J2.0R708, J1.0R956, J1.1R708, J1.0R492, and J1.9R696, were selected as independent variables to construct the optimal subset regression model. the results of the subset analysis are shown in Figure 5a. At the booting stage, nine fractional order differential spectra, J1R756, J1R1140, J2R752, J1R499, J1.9 R654, J1.1R711, J1.8R654, J1.2R654, and J1.7R654, were selected as independent variables to construct the optimal subset regression model and the results of subset analysis are shown in Figure 5b. At the flowering stage, five fractional order differential spectra, J1R754, J1R1127, J1R1309, J1.9R867, and J1.8R820, were selected as independent variables to construct the optimal subset regression model and the results of the subset analysis are shown in Figure 5c. At the filling stage, three fractional order differential spectra, J1R687, J1R538, and J1R752, were selected as independent variables to construct the optimal subset regression model and the results of the subset analysis are shown in Figure 5d.

According to the optimal fractional order differential spectra preferentially selected in different growth stages, the models for LAI estimation at the jointing, booting, flowering, and filling stages were constructed using 75% of sample data based on the optimal subset regression method and 25% of sample data was used for model accuracy validation. The results of *R*^2^, *RMSE*, and *nRMSE* in the modeling and validation of LAI estimation models are shown in Table 1.

As shown in Table 1, the fractional order differential spectrum combined with optimal subset regression for LAI estimation had the best estimation effect at the flowering stage and the worst estimation effect at the booting stage, with *nRMSE* reaching 87.55%. The consistency between the estimated value and the measured value is particularly poor.

#### 3.1.3. Construction of LAI Estimation Model Based on the Support Vector Machine

Select the first 10 fractional differential spectra with a strong correlation with LAI in each growth period as the independent variable, LAI as the dependent variable, and use 75% of the sample data to construct LAI estimation models during the jointing stage, booting stage, flowering stage, and filling stage under the method of SVM. Then, 25% of the sample data was used for accuracy verification. The results of modeling and verification *R*^2^, *RMSE*, and *nRMSE* are shown in Table 2.

As shown in Table 2, when combining fractional order differential spectra with SVM to estimate LAI, the accuracy of LAI estimation at the booting stage and flowering stage was comparable and the estimation at the flowering stage was slightly better than that at the booting stage.

It could be noticed from the comprehensive analysis of LAI estimation results for different growth stages, acquired by using fractional order differential spectra based on optimal subset regression and SVM methods, that at the jointing stage, the overall estimation effect was poor. Although the modeling *R*^2^ of the optimal subset regression model reached 0.72, the *R*^2^ of the model validation was only 0.40, which was a poor estimation effect, while the SVM model had a relatively good estimation effect with *R*^2^ values of 0.65 and 0.60 in the modeling and validation of LAI estimation models, respectively. Overall, the LAI estimation performance at the booting stage was better than that at the jointing stage, the performance of the SVM model estimation was better compared with the optimal subset regression model, and the *R*^2^ in the modeling and validation of the model reached 0.80 and 0.76, respectively. The LAI estimation performance at the flowering stage was the best overall. The estimation performances of optimal subset regression model and the SVM model were comparable and values of *R*^2^ in the modeling and validation of the model reached 0.86, 0.63 and 0.87, 0.57, respectively. Compared with that at the flowering stage, the LAI estimation accuracy was slightly worse at the filling stage. The estimation results of the same optimal subset regression model and SVM model were comparable, with the values *R*^2^ in the modeling and validation of the model reaching 0.84, 0.70 and 0.83, 0.63, respectively.

### 3.2. Estimation of Wheat LAI Based on Continuous Wavelet Transform

#### 3.2.1. Analysis of Correlation between Wavelet Energy Coefficients and LAI

By using the second-order derivative of Gaussian function Mexican Hat as the wavelet basis of the continuous wavelet transform and applying the continuous wavelet transform to the canopy hyperspectral data of wheat at each growth stage, respectively, the wavelet energy coefficients at different scales is obtained. The correlation between LAI and wavelet energy coefficients at different growth periods was analyzed. The correlation graphs and correlation matrices between wavelet energy coefficients and LAI at different fertility periods were plotted, as shown in Figure 6 and Figure 7.

The results of the analysis of Figure 6 and Figure 7 are as follows:

At the jointing stage, the correlation analysis was carried out by using the wavelet energy coefficient and LAI. As shown in Figure 6a, with the increase in the decomposition scale, there is first a rise on the absolute value of the correlation coefficient between wavelet energy coefficient and LAI |ρ|, then a drop. When the decomposition scale was 10, the maximum value of |ρ| was above 0.70, and when the decomposition scale was 7, the maximum value of |ρ| was up to 0.77. With the increase in the decomposition scale, the number of spectral bands passing the 0.01 highly significant level test gradually increased, and when the decomposition scale was 10, the number of bands was up to 1001. The decomposition scale and band where the 10 wavelet energy coefficients with high correlation coefficients were located were C7R725, C2R762, C3R600, C3R417, C6R722, C2R1150, C2R949, C3R766, C3R1276, C5R719, and their correlation matrix with LAI was shown in Figure 7a.

At the booting stage, the correlation analysis was carried out by using the wavelet energy coefficient and LAI. As shown in Figure 6b, with the increase in the decomposition scale, the absolute value of the correlation coefficient between wavelet energy coefficient and LAI |ρ| decreased gradually. When the decomposition scale was 10, the maximum value of |ρ| was above 0.72, and when the decomposition scale was 1, the maximum value of |ρ| could reach 0.83. With the increase in the decomposition scale, the number of spectral bands passing the 0.01 highly significant level test gradually increases; when the decomposition scale was 10, the number of bands can reach up to 1001 bands. The decomposition scales and bands where the 10 wavelet energy coefficients with high correlation coefficients are located are C1R729, C2R727, C3R726, C4R723, C3R1143, C4R1052, C2R755, C3R1055, C5R971, C1R752, and their correlation matrix with LAI is shown in Figure 7b.

At the flowering stage, the correlation analysis was carried out by using the wavelet energy coefficient and LAI. As shown in Figure 6c, with the increase in the decomposition scale, the absolute value of the correlation coefficient between wavelet energy coefficient and LAI |ρ| increased first and then decreased. When the decomposition scale was 10, the maximum value of |ρ| was above 0.70, and when the decomposition scale was 4, the maximum value of |ρ| could reach 0.89. With the increase in the decomposition scale, the number of spectral bands passing the 0.01 highly significant level test gradually increased, and when the decomposition scale was 10, the number of bands could reach 1001. The decomposition scales and bands where the 10 wavelet energy coefficients with high correlation coefficients were located were C4R978, C3R913, C5R985, C4R926, C5R904, C5R990, C6R881, C5R1098, C1R728, C3R1153, and their correlation matrix with LAI was shown in Figure 7c.

At the filling stage, the correlation analysis was carried out by using the wavelet energy coefficient and LAI. As shown in Figure 6d, with the increase in the decomposition scale, the absolute value of the correlation coefficient |ρ| between wavelet energy coefficients and LAI gradually decreased. When the decomposition scale was 10, the maximum value of |ρ| was above 0.80, and when the decomposition scale was 2, the maximum value of |ρ| could reach 0.89. With the increase in the decomposition scale, the number of spectral bands passing the 0.01 highly significant level test gradually increased, and when the decomposition scale was 6, the number of bands was up to 951. The decomposition scales and bands where the 10 wavelet energy coefficients with high correlation coefficients are located are C2R755, C4R782, C1R750, C4R777, C4R790, C3R773, C3R755, C4R768, C5R804, C5R799, and their correlation matrix with LAI was shown in Figure 7d.

#### 3.2.2. Construction and Analysis of LAI Estimation Model Based on Optimal Subset Regression

The results of the optimal subset analysis (shown in Figure 8) in the construction of the LAI estimation model showed that, at the jointing stage, four wavelet energy coefficients, C3R600, C3R417, C6R722, and C5R719, were selected as independent variables to construct the optimal subset regression model, and the results of the subset analysis are shown in Figure 8a. At the booting stage, five wavelet energy coefficients, C1R729, C2R727, C3R726, C4R723, and C1R752, were selected as independent variables to construct the optimal subset regression model, and the results of the subset analysis are shown in Figure 8b. At the flowering stage, seven wavelet energy coefficients, C4R978, C4R926, C5R904, C5R990, C5R1098, C1R728, and C3R1153, were selected as independent variables to construct the optimal subset regression model, and the results of the subset analysis are shown in Figure 8c. At filling stage, the wavelet energy coefficient C2R755 was selected as the independent variable to construct the optimal subset regression model, and the results of the subset analysis are shown in Figure 8d.

According to the optimal fractional differential spectrum selected in different growth stages, the models for LAI estimation at the jointing, booting, flowering, and filling stages were constructed using 75% of sample data based on the optimal subset regression method and 25% of sample data were used for model accuracy validation. The results of *R*^2^, *RMSE*, and *nRMSE* in the modeling and validation of LAI estimation models are shown in Table 3.

As shown in Table 3, when combining wavelet energy coefficients with optimal subset regression to estimate LAI, the accuracy at the flowering stage was the highest; its modeling accuracy was (*R*^2^ = 0.87, *RMSE* = 0.43µg/cm^2^, *nRMSE* = 12.84%) and the verification accuracy was (*R*^2^ = 0.71, *RMSE* = 0.66µg/cm^2^, *nRMSE* = 19.15%).

#### 3.2.3. Construction and Analysis of LAI Estimation Model Based on the Support Vector Machine

Select the first 10 fractional differential spectra with a strong correlation with LAI in each growth period as the independent variable, LAI as the dependent variable, and use 75% of the sample data to construct LAI estimation models during the jointing stage, booting stage, flowering stage, and filling stage under the method of SVM. Then, 25% of the sample data was used for accuracy verification. The results of modeling and verification *R*^2^, *RMSE*, and *nRMSE* are shown in Table 4.

Table 4 showed that, when combining wavelet energy coefficients with SVM to estimate the LAI number, the accuracy at the flowering stage and filling stage were comparable and their estimation effect was better than that at the jointing stage and booting stage. In addition, their modeling *R*^2^ was above 0.85, validation *R*^2^ was greater than 0.60, and the modeled *nRMSE* was less than 20%.

It could be noticed from the comprehensive analysis of LAI estimation results for different growth stages, acquired by using wavelet energy coefficient based on optimal subset regression and SVM methods, that, compared with other growth stages, the overall estimation effect of LAI at the jointing stage was poor, and the estimation effect of optimal subset regression model and SVM model was equivalent, with the modeling and verification *R*^2^ reaching 0.73, 0.59 and 0.69, 0.74, respectively. At the booting stage, the LAI estimation effect was better than that in jointing stage. Compared with SVM model, the optimal subset regression model had better estimation effect, with the modeling and verification *R*^2^ of the model reaching 0.81 and 0.56, respectively. At the flowering and filling stages, LAI estimation effect was equivalent and the estimation effect of optimal subset regression model and SVM model was also equivalent; the modeling *R*^2^ of the model reached 0.87, 0.84 and 0.87, 0.90, respectively, and the verification *R*^2^ reached 0.71, 0.77 and 0.65, 0.63, respectively.

## 4. Discussion

By comprehensively analyzing the estimation results of wheat LAI at four growth stages using fractional differential spectrum and wavelet energy coefficient based on optimal subset regression and SVM method, it was noticed that the maximum values of *R*^2^ in modeling and verification of the estimation model were 0.90 and 0.77, respectively, and the average values of *R*^2^ in modeling and verification were 0.79 and 0.64, respectively, indicating generally that the LAI estimation performance was good, which is mainly due to the following reasons. First, the data source is hyperspectral data, which have the advantages of high spectral resolution and rich spectral information of features, able to be used to analyze the spectral characteristics of features from multiple angles and directions and comprehensively express the detailed information of crops, and are suitable for remote sensing estimation of crop phenotype parameters. Second, the difficulty of crop LAI estimation using hyperspectral data lies in the accurate extraction of LAI-sensitive spectral information from hyperspectral data; however, the data acquisition process was affected by factors such as environment and background, resulting in noise in the original spectral data that affected the extraction of sensitive information. After fractional order differentiation and wavelet transform processing, the noise effect can be effectively eliminated, the spectral information can be refined, and the crop LAI estimation effect can be effectively improved, which is consistent with the conclusions of research by Li Changchun et al. [22], Fang Shenghui et al. [34], and Yao Shengnan et al. [35].

The result of comprehensive analysis on the effect of LAI estimation at different growth stages showed that, at the jointing stage, the maximum values of *R*^2^ in modeling and verification were only 0.73 and 0.74, which was relatively poor compared with that at other growth stages. The values of *R*^2^ in the modeling and validation of the LAI estimation model at the flowering stage and the filling stage were slightly different, with the average values of 0.87, 0.85 and 0.64, 0.68, respectively. The modeled *nRMSE* was slightly different, with the average values of 13.92 and 18.10%, respectively, both of which were less than 20%; however, at the filling stage, the maximum value of verified *nRMSE* was 41.69% and the average value was 35.98%, while at the flowering stage, the maximum value of verified *nRMSE* was only 29.93% and the average value was 24.66%, indicating that the LAI estimation effect at the flowering stage was slightly better than that at the filling stage, which is mainly because wheat growth reaches its peak at the flowering stage and then LAI starts to decrease. In addition, LAI is relatively sensitive to the canopy spectrum.

The result of comprehensive analysis on the LAI estimation effects of different methods at different growth stages showed that, at the jointing stage, the estimation effect of the optimal subset regression model based on wavelet energy coefficient was equivalent to that of the SVM model, and the *R*^2^ values in modeling and verification were 0.73, 0.59 and 0.69, 0.74, respectively. At the booting stage, the LAI estimation accuracy of the SVM model based on fractional differential spectrum was the highest, and the *R*^2^ values in modeling and verification reached 0.80 and 0.76, respectively. At the flowering stage, the LAI estimation accuracy of the optimal subset regression model based on wavelet energy coefficient was the highest, and the *R*^2^ values in the modeling and verification were 0.87 and 0.71, respectively. At the filling stage, the LAI estimation accuracy of the optimal subset regression model based on wavelet energy coefficient was the highest, and the *R*^2^ values in the modeling and verification reached 0.84 and 0.77, respectively. It provided the best scheme for wheat LAI estimation at different growth stages.

For four growth periods, the effect of LAI estimation by optimal subset regression models and SVM models based on fractional order differential spectra and wavelet energy coefficients was analyzed. Compared with fractional differential spectrum, the optimal subset regression model and SVM model based on wavelet energy coefficient have better LAI estimation effect; the maximum values of *R*^2^ in modeling and verification were 0.90 and 0.77, respectively, and the average values were 0.81 and 0.66, respectively. This was mainly because, as a new spectral processing method, wavelet transform can effectively reduce noise and decompose spectral data, mine spectral hidden information, extract more sensitive information, and effectively improve the accuracy and generalization ability of the model, which was consistent with the research conclusions of Ebrahimi et al. [23] and Cheng et al. [36].

## 5. Conclusions

In this paper, the collected hyperspectral data of wheat canopy at different growth stages were processed by fractional order differentiation and continuous wavelet transform. According to the correlation analysis results, the fractional order differential spectrum and wavelet energy coefficients with strong correlation were selected. Combined with the optimal subset regression and SVM method, LAI estimation models of wheat at different growth stages were constructed, respectively. Based on the results of modeling and validation accuracy assessment to screen the optimal estimation model of LAI at each growth stage and the fertility period with the best LAI estimation, this study can provide a theoretical and technical reference for LAI estimation of crops based on remote sensing technology. However, the LAI estimation methods used in the paper have some limitations and need to be further improved in the following two aspects.

The optimal subset regression and support vector machine algorithms used in the paper are both empirical regression models, which ignore the radiative transfer processes of vegetation canopy and atmosphere. The next study needs to parameterize the remote sensing radiative transfer processes to further improve the stability of the model.

Due to the limitation of the experimental conditions, only the sample data of one experimental area were used in this study, and the influence of experimental data by locality was not considered. The subsequent study can verify the prediction effect of the model by increasing the sample size, crop species, and sample years, which is very meaningful for the further transfer of the model to practical applications.

## Figures and Tables

**Figure 1 sensors-21-08497-f001:**
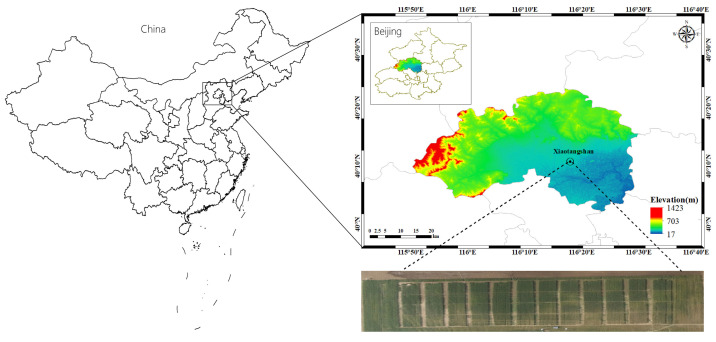
Schematic diagram of the study area and experimental design.

**Figure 2 sensors-21-08497-f002:**
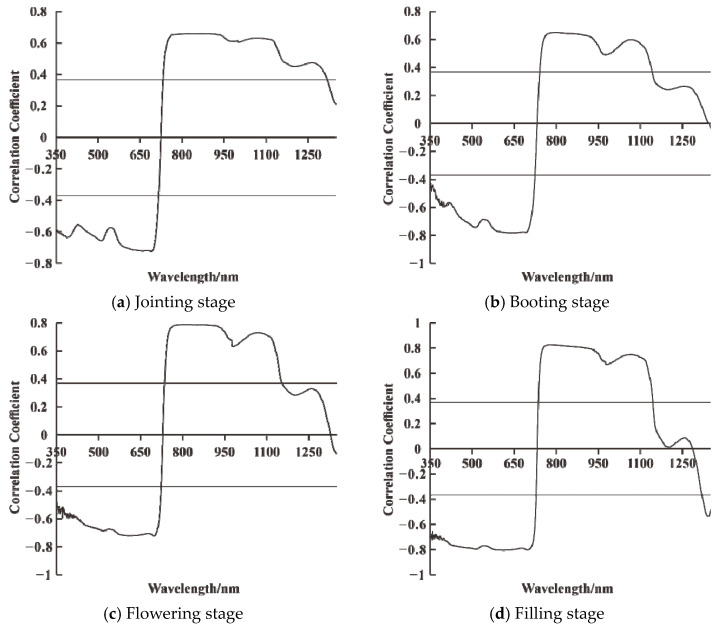
Correlation analysis of original spectrum and LAI at different growth stages.

**Figure 3 sensors-21-08497-f003:**
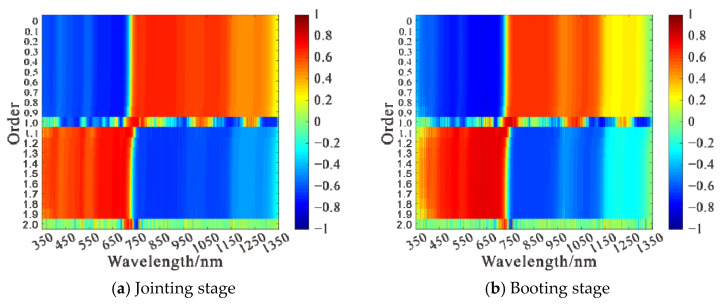
Correlation analysis of fractional differentiation spectrum and leaf area index at different growth stages.

**Figure 4 sensors-21-08497-f004:**
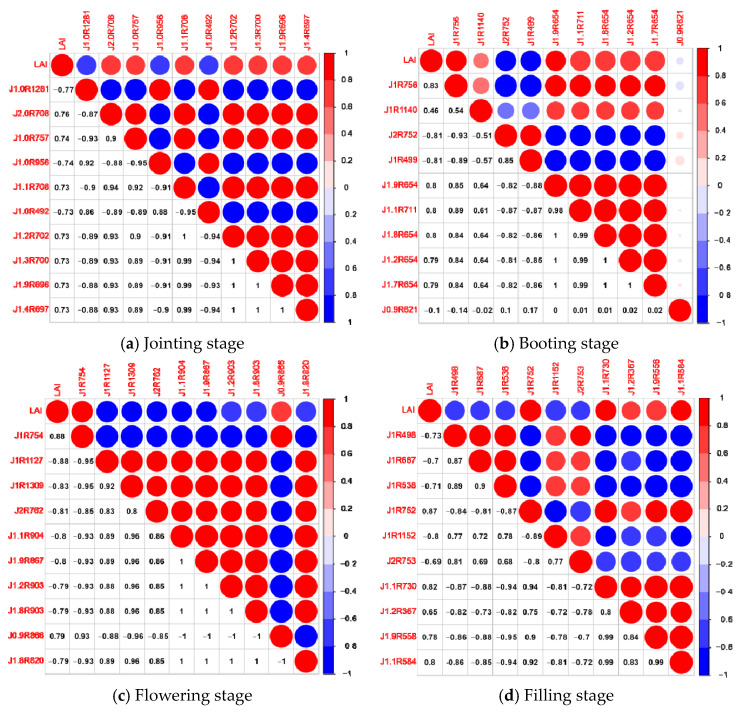
Correlation matrix diagram of selected fractional differentiation spectrum and leaf area index at different growth stages.

**Figure 5 sensors-21-08497-f005:**
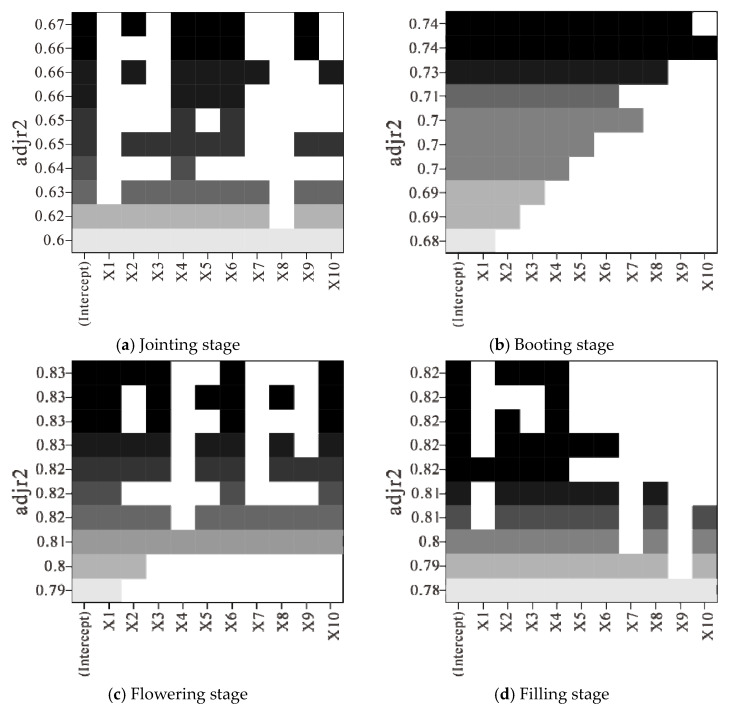
Optimal subset analysis of selected fractional differentiation spectrum for estimating LAI.

**Figure 6 sensors-21-08497-f006:**
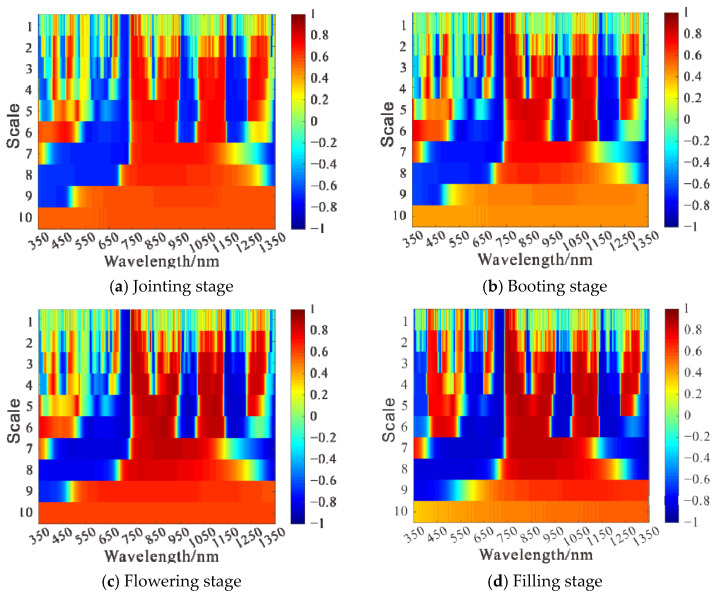
Correlation analysis of wavelet energy coefficient and leaf area index at different growth stages.

**Figure 7 sensors-21-08497-f007:**
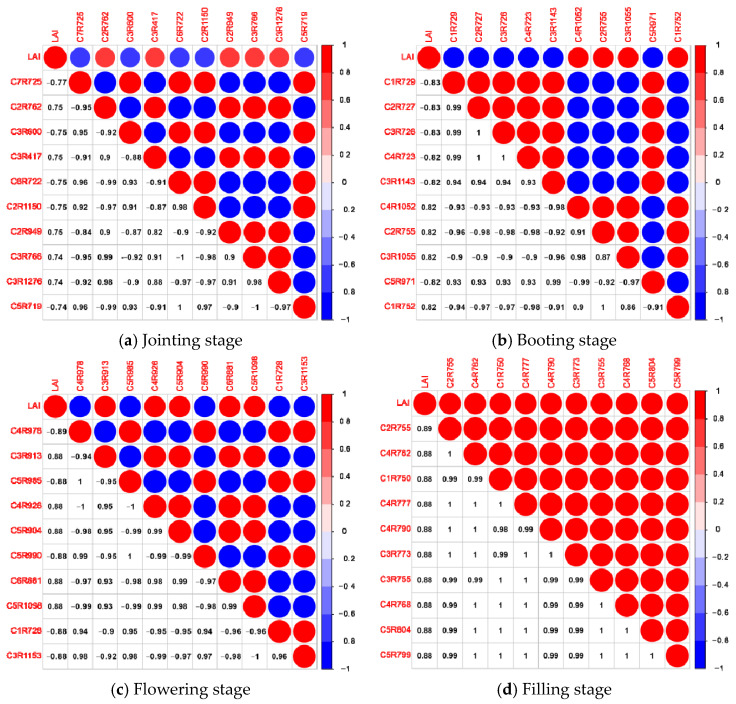
Correlation analysis of selected wavelet energy coefficient and leaf area index at different growth stages.

**Figure 8 sensors-21-08497-f008:**
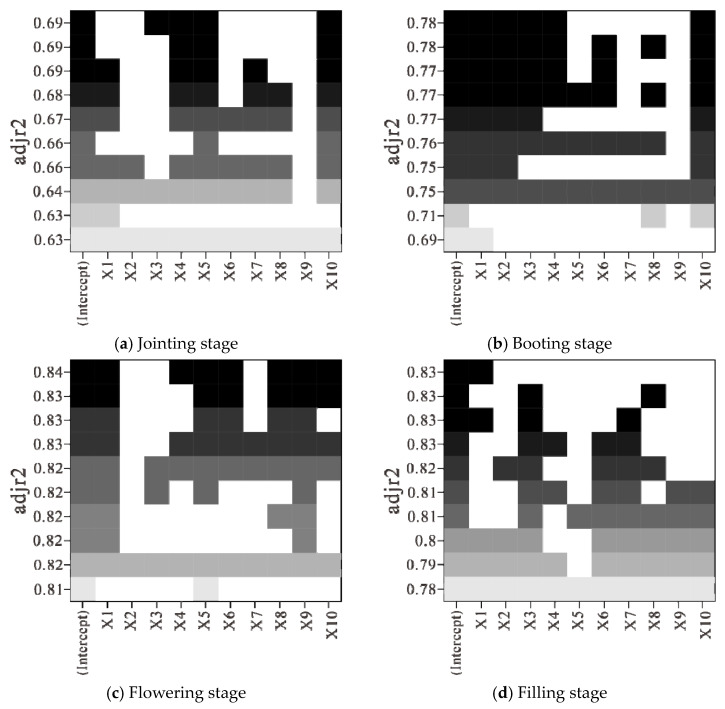
Optimal subset analysis of wavelet energy coefficient for estimating leaf area index at different growth stages.

**Table 1 sensors-21-08497-t001:** Optimal subset regression modeling results of LAI estimation based on fractional differentiation spectrum at different growth stages.

Growth Stages	Modeling Accuracy			Verification Accuracy		
	*R* ^2^	*RMSE* (µg/cm^2^)	*nRMSE* (%)	*R* ^2^	*RMSE* (µg/cm^2^)	*nRMSE* (%)
Jointing stage	0.72	0.47	12.65%	0.40	0.90	26.30%
Booting stage	0.67	1.75	61.86%	0.56	2.06	87.55%
Flowering stage	0.86	0.45	13.43%	0.63	0.74	21.61%
Filling stage	0.84	0.31	19.78%	0.70	0.62	41.69%

**Table 2 sensors-21-08497-t002:** Estimation of leaf area index based on fractional differentiation spectrum at different growth stages and modeling results of support vector machine.

Growth Stages	Modeling Accuracy			Verification Accuracy		
	*R* ^2^	*RMSE* (µg/cm^2^)	*nRMSE* (%)	*R* ^2^	*RMSE* (µg/cm^2^)	*nRMSE* (%)
Jointing stage	0.65	0.56	15.12%	0.65	0.60	17.09%
Booting stage	0.80	0.83	20.01%	0.76	0.82	23.32%
Flowering stage	0.87	0.47	13.94%	0.57	0.80	28.37%
Filling stage	0.83	0.38	21.24%	0.63	0.51	36.76%

**Table 3 sensors-21-08497-t003:** Optimal subset regression modeling results of leaf area index estimation based on wavelet energy coefficient at different growth stages.

Growth Period	Modeling Accuracy			Verification Accuracy		
	*R* ^2^	*RMSE* (µg/cm^2^)	*nRMSE* (%)	*R* ^2^	*RMSE* (µg/cm^2^)	*nRMSE* (%)
Jointing stage	0.73	0.46	12.40%	0.59	0.71	19.47%
Booting stage	0.81	0.78	18.46%	0.56	0.95	23.64%
Flowering stage	0.87	0.43	12.84%	0.71	0.66	19.15%
Filling stage	0.84	0.31	19.91%	0.77	0.58	38.55%

**Table 4 sensors-21-08497-t004:** Estimation of leaf area index based on wavelet energy coefficient at different growth stages and modeling results of support vector machine.

Growth Period	Modeling Accuracy			Verification Accuracy		
	*R* ^2^	*RMSE* (µg/cm^2^)	*nRMSE* (%)	*R* ^2^	*RMSE* (µg/cm^2^)	*nRMSE* (%)
Jointing stage	0.69	0.52	14.37%	0.74	0.58	16.83%
Booting stage	0.76	0.88	20.80%	0.63	0.87	23.40%
Flowering stage	0.87	0.42	15.47%	0.65	0.73	29.93%
Filling stage	0.90	0.39	11.47%	0.63	0.76	26.91%

## Data Availability

Since the datasets were acquired through field collection by the Beijing Research Center for Information Technology in Agriculture, all data cannot be shared due to legal, ethical, and privacy restrictions.

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
