# Peer review of "Hyperspectral Estimation of Winter Wheat Leaf Area Index Based on Continuous Wavelet Transform and Fractional Order Differentiation"

_sensors, 2021, doi:10.3390/s21248497_

Round 1
Reviewer 1 Report
Referee's Report
TITLE: Hyperspectral Estimation of Winter Wheat Leaf Area Index Based on Continuous Wavelet Transform and Fractional Order Differentiation
JOURNAL: Sensors, MDPI
Summary and Recommendation:
The presented idea is interesting and generally can be applied for further research. However, this paper has many grammatical and human errors. I suggest publishing it after carefully addressing all the highlighted comments and suggestions.
- TITLE: In my opinion, if the authors generalized the integer-order model to fractional-order, I suggest rephrasing the title to:
``A Fractional Order Hyperspectral Estimation of Winter Wheat Leaf Area Index Based on Continuous Wavelet Transform”
- ABSTRACT:
- Line 14-16, need to be rephrased and arranged because the sentence meaning is not understandable.
- There are a lot of mismatches in the Abstract Section, I recommend the authors revise the abstract with care.
- INTRODUCTION:
- The original contribution of fractional order is not well explained. A more discussion is needed related to the motivation of fractional order.
- The readability and presentation of the study should be further improved. The paper suffers from language problems. The paper should be proofread by a native speaker or a proofreading agent.
- Model Accuracy Evaluation:
- Punctuation marks are missing, see for example equations 4,5, and 6 respectively.
- CONCLUSION:
- Please see lines 593 and 603, you must be consistent throughout the paper.
Reviewer 2 Report
The research work is good and can be accepted after minor modifications.
- The observations of weather ob crop canopy should be analysed,
- What are the factors effecting the growth of crop ?
- The limitations of research work should also be explained.
Reviewer 3 Report
The article is written correctly, as required by the journal. The introduction has an appropriate number of references to previous research and clearly defines the purpose and scope of the work. A very serious drawback of the article is its language. The reviewer found a lot of errors, and therefore suggests that the article be subject to a linguistic proofreading. Additionally, some expressions in the Abstract are completely incomprehensible. This part of the article needs clarification. The main content-related remark by the Reviewer is the lack of a critical assessment of the constructed algorithms. The authors do not clearly indicate the limitations of the method used to evaluate the LAI. This remark should be addressed in Conclusions.The presented comments do not reduce the high scientific value of the article, therefore I refer the article to the next stages of evaluation after making minor revision.
